# Human Papilloma Virus Vaccination in Patients with Rheumatic Diseases in France: A Study of Vaccination Coverage and Drivers of Vaccination

**DOI:** 10.3390/jcm11144137

**Published:** 2022-07-16

**Authors:** Emmanuelle David, Pascal Roy, Alexandre Belot, Pierre Quartier, Brigitte Bader Meunier, Florence A. Aeschlimann, Jean-Christophe Lega, Isabelle Durieu, Christine Rousset-Jablonski

**Affiliations:** 1Service de Médecine Interne et Pathologies Vasculaires, Hospices Civils de Lyon, Hôpital Lyon Sud 165 Chemindu Grand Revoyet, 69495 Pierre-Bénite, France; jean-christophe.lega@chu-lyon.fr (J.-C.L.); isabelle.durieu@chu-lyon.fr (I.D.); christine.rousset-jablonski@chu-lyon.fr (C.R.-J.); 2Laboratoire de Biométrie et Biologie Evolutive, Equipe Biostatistique-Santé, CNRS, 43 Bd du 11 Novembre 1918, CEDEX, 69622 Villeurbanne, France; pascal.roy@chu-lyon.fr; 3Service de Biostatistique et Bioinformatique, Hospices Civils de Lyon, 162, Avenue Lacassagne, CEDEX 03, 69424 Lyon, France; 4Reference Centre for Rare Pediatric Rheumatic Diseases RAISE, Service de Néphrologie, Rhumatologie, Dermatologie Pédiatriques, Hôpital Femme-Mère-Enfant, Hospices Civils de Lyon, 59 Bd Pinel, 69500 Bron, France; alexandre.belot@chu-lyon.fr; 5Reference Centre for Rare Pediatric Rheumatic Diseases RAISE, Service D’immunologie, Hématologie et Rhumatologie Pédiatrique, Hôpital Necker, APHP, Institut IMAGINE, 149 Rue de Sèvres, 75015 Paris, France; pierre.quartier@aphp.fr (P.Q.); brigitte.bader-meunier@aphp.fr (B.B.M.); florence.aeschlimann@aphp.fr (F.A.A.); 6Department of Paediatric Immunology, Hematology and Rheumatology and Institut IMAGINE, Necker-Enfants Malades Hospital, Université Paris-Cité, 75006 Paris, France; 7INSERM U1290 RESHAPE, Domaine Rockefeller 8 avenue Rockefeller 69373 Lyon CEDEX 08, Département de Formation et de Recherche en Biologie Humaine, Université Claude Bernard Lyon 1, 69100 Lyon, France; 8Centre Léon-Bérard, 28 Prom. Léa et Napoléon Bullukian, 69008 Lyon, France

**Keywords:** systemic erythematosus lupus, juvenile idiopathic arthritis, vaccination, human papillomavirus, barriers

## Abstract

Objectives: To describe human papillomavirus (HPV) vaccination practices in adolescent girls with systemic lupus erythematosus (SLE) and juvenile idiopathic arthritis (JIA) and to identify barriers to and motivators for vaccination. Methods: Cross-sectional, multicenter study on girls aged 9 to 19 years and their accompanying adults. The measurement criteria were the proportion of girls who were vaccinated against HPV, compliance with the vaccination schedule, factors associated with vaccination, and reasons for vaccination and non-vaccination through a self-administered questionnaire. Results: Seventy-one patients (16 with SLE and 55 with JIA) were included with a mean age of 13 years old (rank 11–18). According to parental questioning, 39% of patients were vaccinated against HPV or in progress (44% and 38% of SLE and JIA, respectively). This rate was 82% for the 22 patients ≥ 15 years of age. The vaccine was administered as often by a general practitioner (39%) as by a hospital pediatrician (also 39%). Two factors were significantly associated with vaccination: Older age (OR 53.68, 95% CI 5.85–429.29, *p* < 0.001) and previous hepatitis B vaccination (OR 4.97, 95% CI 1.03–24.01, *p* = 0.040). Recommendation of the vaccine by a health professional and fear of HPV-related diseases were the main facilitators. Lack of knowledge about the vaccine, lack of recommendation by a health professional, and fear of vaccine side effects were the main barriers. Conclusions: HPV vaccination coverage remains insufficient among patients with autoimmune disease. Education and awareness of health professionals about HPV infections are crucial elements in vaccine acceptance.

## 1. Introduction

Human papillomavirus (HPV) infection is one of the most widespread sexually transmitted infections. HPV is the main causative agent in the development of precancerous and cancerous lesions of the cervix, mainly related to HPV 16 and 18 [1]. Underlying conditions, such as immunosuppression, favor the persistence of intraepithelial lesions and the progression to an aggressive form. Patients with autoimmune diseases, particularly systemic lupus erythematosus (SLE) [2] and rheumatoid arthritis [3] or even patients with chronic inflammatory bowel [4], can be prone to HPV infection and subsequent cervical dysplasia [5]. Some studies confirm more HPV infections ranging from 20.2% (versus 7.3% in the control population) to 80.7% (versus 35.7% in control population) [2,6]. The incidence of HPV-related dysplasic lesions in lupus patients is significantly increased to 8.66 times of the general population in a meta-analysis (OR 8.66–95% CI 3.75 to 20.00) [7]. Some authors measure up to 3.5% of high-grade lesions on smear in a group of 85 lupus patients against 0.5% in healthy controls [8]. One study found a higher frequency of HPV infection among patients with juvenile idiopathic arthritis (JIA) without significance [9]. The cause of this interaction is not yet known. Some immunosuppression treatments, and even more with long-term use, could promote the development of uterine and cervical lesions [6]. Dysregulation of the immune system [10] or the disease itself [2] has also been suggested to be an associated factor. Although SLE patients seem to have more premalignant lesions than healthy women, the prevalence of more cervical cancer for these patients than for the healthy population remains to be discussed [11,12]. When administered before the beginning of sexual activity, the effectiveness of vaccination to prevent infection by the HPV included in the vaccine is close to 100%. In France, vaccination against HPV is recommended for girls and boys aged 11 to 14 years old. HPV vaccination can also be proposed earlier (from the age of 9 years old) in cases of immunosuppressive treatment [13]. In addition, as part of the vaccination catch-up, vaccination is proposed for young women and men between 15 and 19 years of age who remain unvaccinated. At the time of the study, this recommendation was limited to girls, and three types of vaccines were available: Bivalent, quadrivalent, and nonvalent. Several studies have demonstrated the efficacy and safety of the vaccine, including in immunocompromised patients and those with lupus [14]. However, the acceptability of the HPV vaccine remains insufficient, and vaccination coverage has remained low among French girls since its introduction in 2007 (24% in 2018) [15,16]. Vaccine coverage for HPV is not known for patients with pediatric lupus and juvenile idiopathic arthritis in France. Therefore, in this study, we first aimed to estimate HPV vaccination uptake among girls with SLE and JIA who were eligible to be vaccinated. The second objective was to investigate the motivators for, barriers to, and factors associated with vaccination.

## 2. Material and Methods

We conducted a multicentric, cross-sectional study among young girls and their accompanying adults between January 2020 and July 2021 in 2 French tertiary centers: Hôpital Femme-Mère Enfant, Lyon (Auvergne Rhône-Alpes) and Hôpital Necker-Enfants-Malades, Paris (Ile-de France).

### 2.1. Study Population

All patients with a confirmed diagnosis of SLE or JIA, aged 9–19 years old, who presented for a follow-up visit at one of the participating centers accompanied by an adult were eligible to participate. The exclusion criteria were refusal to answer the questionnaire by the patient or her parent.

### 2.2. Data Collection

The patient and her accompanying adult completed a self-questionnaire. The first part was completed by the girl, and the second part was completed by the accompanying adult. The questionnaire included closed- and open-ended questions. The items completed by the girl included demographic information (age, educational level) and HPV vaccination status. For the accompanying adult, the following information was collected: Demographic data (age, educational level, relationship with the patient: Father, mother, other), vaccination status for themselves against hepatitis B, and vaccination status for their child against HPV, pneumococcus, flu, meningococcus C, hepatitis B. Information regarding current pathology (JIA or SLE) and date of diagnosis was collected. Treatments received during follow-up were also collected and included hydroxychloroquine, nonsteroidal anti-inflammatory agents, corticoids, anti-interleukin 1, anti-interleukin 6, methotrexate, sulfasalazine, leflunomide, azathioprine, anti-TNF-alpha, CTLA-4, cyclosporine, mycophenolate mofetil, cyclophosphamide, rituximab, thalidomide, belimumab, or other (open-ended answer).

For vaccinated patients, additional information was collected: Prescriber (general practitioner, liberal pediatrician, hospital-based pediatrician, gynecologist), type of vaccine (bivalent, quadrivalent, nonvalent), and numbers and dates of injections. The reasons for vaccination were collected from the patients and their parents through closed-ended items (several answers possible): Fear of HPV-related diseases, health care provider (HCP) or relative guidance on HPV vaccination, sensitization following a relative’s disease, no specific opinion, or other (open-ended).

For unvaccinated patients, reasons for non-vaccination were collected from the girl and her accompanying adult. The following reasons were proposed (several answers possible): Lack of knowledge about the HPV vaccine, vaccine not proposed by a prescriber, concerns about potential side effects, uncertain opinion because the vaccine had been introduced too recently, perceived uselessness, perceived inefficiency, not recommended by a relative, not recommended by a HCP, price/refund difficulty, past bad experience with a vaccine, fear of a disease flare, vaccination planned later, or other (open-ended).

Educational level was assessed based on the International Standard Classification of Education (ISCED) [17]. Respondents were categorized into 3 groups: low (ISCED level 0–2), medium (ISCED level 3–4), and high educational level (ISCED ≥ 5). The level of immunosuppression was categorized into 3 groups according to the treatments received: Low (hydroxychloroquine, AINS, thalidomide), mild (corticoids, methotrexate, sulfasalazine, leflunomide, azathioprine), or high (anti-interleukin 1, anti-interleukin 6, anti-TNF-alpha, CTLA-4, cyclosporine, mycophenolate mofetil, cyclophosphamide, rituximab, belimumab).

If the mother was the accompanying adult, she was asked about cervical cancer prevention practices, including participation in cervical cancer screening and previous abnormal Pap smear results.

All of the answered questionnaires were returned to the coordinating center (Lyon) for data recording and processing.

### 2.3. Outcome Measures

The main outcome measure was the proportion of girls who received or were receiving HPV vaccination in the whole included population and in each age category (9–10 years old, 11–14 years old, and 15 years old and older). In the event of discordance between the girl’s answer and that of the accompanying person, the accompanying person’s answer was considered in the analysis. Other outcome measures were as follows: The proportion of vaccinated girls who complied with the vaccination schedule according to the standard vaccination schedule in France at the time of the study (Appendix A) [18], the type of vaccine (bivalent, quadrivalent, nonvalent), and factors associated with vaccination and non-vaccination. A patient was compliant if all doses had been administered with the correct spacings between doses according to her age. If data were missing concerning an injection date, then data for compliance with the vaccination schedule were considered to be missing for the patient concerned. Notably, since 31 December 2020, the quadrivalent human papillomavirus (HPV) vaccine Gardasil is no longer marketed in France.

### 2.4. Ethical Approval

The study was approved by the Comité de Protection des Personnes du Sud-Ouest et Outre-Mer (3 July 2019) and was registered at clinicaltrials.gov (NCT04180228).

### 2.5. Statistical Analysis

Medians and the 1st and 3rd quartiles were reported for quantitative variables. Nonparametric Wilcoxon rank sum tests were performed to compare distributions of quantitative covariates between groups. Frequencies were reported for categorical variables. The associations between HPV vaccination status and exploratory covariates were analyzed by fitting unconditional logistic regression models. Odds ratios were presented with corresponding 95% confidence intervals (95% CI). Nested logistic regression linear models were compared using likelihood ratio tests. Fisher’s exact test was performed in the case of small numbers. For all statistical tests, *p* values (two-tailed) less than 5% were considered significant. Variables significantly associated with HPV status in univariate analyses were considered for multivariate modeling.

## 3. Results

### 3.1. Study Population

The questionnaire was offered to 82 girls consulting at participating centers between January 2020 and July 2021. A total of 71 patients (16 SLE and 55 JIA patients) were included in the analysis. Flowchart of the study participants is presented in Figure 1.

The main characteristics of the patients with completed data and their accompanying adults are detailed in Table 1.

### 3.2. HPV Vaccination Status

A total of 26 (37%; 95% CI% (25%; 49%)) patients declared that they had received HPV vaccinations. Twenty-eight accompanying adults (39%; 95% CI% (28%; 52%)) reported that their daughter had been or was being vaccinated. Among SLE patients, 7 of 16 received vaccination (44%), and 21 of 55 JIA patients (38%) were vaccinated. The mean age at vaccination (first injection) was 13.6 years (SD 1.6). The age at first injection was unknown in 11 of 28 patients. Of the 17 remaining vaccinated patients, 12 (71%) received the first injection between 11 and 14 years of age, and 5 patients (29%) received the first injection after 15 years of age. Eighteen of the 29 girls aged 15 years old or older (62%) and 10 of the 41 girls aged 11–14 years old (24%) had been vaccinated.

Among the 43 nonvaccinated patients, 14 (33%) parents were planning to vaccinate their daughters later.

### 3.3. Vaccinated Patients

The characteristics of the 28 vaccinated patients (44% with SLE, 38% with JIA, respectively) are detailed in Table 2. Patients had mainly received the quadrivalent vaccine. Four of 28 patients did not have adequate vaccination schedules. Vaccines were mainly prescribed by hospital pediatricians or general practitioners.

### 3.4. Motivations for and Barriers to Vaccination

The main reasons given by accompanying adults and girls for vaccination are described in Figure 2A. The principal motivation factor reported by participants for initiating vaccination was a recommendation from an HCP (68% of adults (19/28) and 58% of girls (15/26)), followed by fear of HPV-related diseases (54% of adults (15/28) and 38% of girls (10/26)). Sensitization because of a relative’s disease was cited only by the adults (11%, 3/28). Only 2 parents (7%) said they had no specific opinion about the vaccine, compared to 5 girls (19%).

The main justifications declared by the accompanying parents and their daughters for non-vaccination are described in Figure 2B. The main barriers identified by the participants were a lack of knowledge about the vaccine for 8 of 43 parents (19%) and 16 of 45 girls (36%) and lack of proposal by an HCP (13/43 parents and 11/45 girls). No girls or adults reported any price, refund difficulty or lack of efficacy as a reason for non-vaccination. Few respondents were afraid of a disease flare or were advised against the vaccine by a relative or an HCP. An uncertain opinion because the vaccine had been introduced too recently or concerns about side effects were mainly cited by parents (8/43 (19%) and 10/43 (23%), respectively) but not as much by girls (2/45 (4%) and 1/45 (2%), respectively). Many parents planned to vaccinate their daughters later (14/43 (33%)), and the daughters’ answers were concordant (13/45 (29%)).

### 3.5. Comparison of Vaccinated and Nonvaccinated Patients

A comparison between vaccinated and nonvaccinated patients and their accompanying adults is detailed in Table 3. Non-vaccinated patients were significantly younger than vaccinated patients (median age 12 vs. 15; *p* < 0.001), whereas the age of accompanying adults did not influence vaccination (*p* = 0.453). Vaccination was less frequent among girls attending the Lyon center than among girls attending the Parisien center (OR = 0.33, 95% CI 0.12–0.89, *p* = 0.025). Vaccinated girls were significantly more likely to be vaccinated for hepatitis B (OR = 4.63, 95% CI 1.38–15.56, *p* = 0.009), as were their parents (parents (OR = 6.48, 95% CI 1.30–32.17, *p* = 0.008). There was no association with other usual vaccines proposed for patients suffering from autoimmune conditions, such as flu, pneumococcus, or meningococcus C (*p* > 0.05). We did not observe any difference in the proportion of HPV vaccinations among lupus or JIA patients (*p* = 0.698). Multivariate analysis (Appendix A) confirmed this result with an association of HPV vaccination status with anteriority of patient hepatitis B vaccination when adjusted on age group (OR 4.97, 95% CI 1.03–24.01, *p* = 0.040), or when adjusted on clinical center (OR 4.01, 95% CI 1.15–14.00, *p* = 0.040). We did not find any associations of parental educational level, maternal gynecological history (regular cervical screening, previous abnormal cervical smear), duration of disease evolution, or immunosuppression level.

## 4. Discussion

The proportion of girls undergoing or having been vaccinated against HPV was 39% (95%CI 28–52%). Almost half of the lupus patients and more than one-third of the JIA patients were vaccinated. Among girls aged 15 years old and older (assumed to have been previously vaccinated), this proportion was 62%. In addition, 14 parents planned to vaccinate their daughters later. Factors associated with vaccination were older age and vaccination of the child for hepatitis B. Vaccination coverage varied by health center. Insufficient knowledge about the HPV vaccine, concerns about potential side effects, concerns about the novelty of the vaccine, and lack of proposition from an HCP were the main barriers to vaccination.

The prevalence of HPV vaccination in girls with autoimmune conditions is insufficient but remains higher than that described in the French general population [19]. However, vaccination in the general population appears to be increasing slightly, with 32.7% of 16-year-old girls vaccinated in 2020 versus 23.7% in 2018. Among lupus patients, 43.7% received the HPV vaccine in our study. Our result is somewhat divergent from the few studies investigating HPV vaccination coverage in lupus patients. In a cohort of 1349 women with SLE, of 237 eligible for vaccination, only 4.6% were vaccinated [20]. In another series of 5642 patients with chronic inflammatory disease, only 21% had received at least one dose of HPV vaccine, compared with 23% in the general population, without a significant difference. In the subgroup, among 299 individuals with SLE, 32 patients (10.7%) received at least one dose of vaccine [21]. Our results are probably explained in part by a small number of subjects and a recruitment bias by the center effect. In addition, the majority of participants were between the ages of 11 and 14, and a larger age sample would be more representative. A positive impact of recent prevention campaigns in favor of the vaccine and the general national trend of increasing vaccinations must also be taken into account. Despite an increased risk of vaccine-preventable infection, adolescents with autoimmune disease vaccination coverage for mandatory vaccine levels remain suboptimal [22,23], and similar results for specifically recommended vaccines, such as influenza or pneumococcal, are found [24]. Similar insufficient vaccination coverage is also described in children with other chronic diseases despite the known increased risks of infectious diseases [25,26].

We assessed factors independently associated with HPV vaccination. We found a positive association with hepatitis B vaccination, as found elsewhere [27,28]. This result could reflect a generally positive attitude towards vaccination, especially in France, where the controversy over hepatitis B vaccination and the risk of multiple sclerosis remains relevant. As found in the literature [28,29], we thought that mothers’ attitudes towards cervical cancer prevention or a history of abnormal Pap smears might influence their daughters’ vaccination, but this correlation was not found in our study. We did not find any association with the level of immunosuppression or the duration of the disease. Parental education level did not appear to influence vaccination status herein, although previous studies have described lower acceptance of HPV vaccination among parents with higher education levels [30].

The main motivating factor for vaccination in our study was the recommendation of an HCP. Strong provider recommendation is also a key determinant of HPV vaccine uptake in the general population [31,32]. This point is also important for adolescents with chronic medical conditions (CMCs), but many providers fail to recommend or discuss vaccines with these high-risk patients [33,34]. The lack of a proposal was one of the main reasons for not vaccinating in our study since 30% of parents stated that they had never received a proposal for an HPV vaccine for their child. Of one hundred US pediatricians surveyed about their HPV vaccination practices, only 50% sometimes or always recommended HPV vaccination to their patients being followed for chronic diseases [35]. In our cohort, 39% of prescribers were general practitioners. Vaccination is often relegated to the general practitioner, and primary prevention procedures are too often neglected by specialist physicians [24,35]. The main barriers cited by HCPs are a lack of knowledge [35], time constraints on discussing HPV vaccination [36], and discomfort with discussing sexual health [35]. By avoiding the subject of vaccination, HCPs unintentionally convey a message of distrust of the vaccine [37], although their influence on the parental decision is crucial [38]. However, many adolescents with CMC consider the specialist to be their main provider [39,40]. The education of professionals about the vaccine, therefore, seems essential. Lack of efficacy was not cited as a limiting factor to vaccination. In fact, vaccine immunogenicity studies have shown effective vaccine protection despite a decrease in seroconversion compared to control populations [14,41].

Reasons for parents’ reticence to vaccinate were similar to those described in other cohorts of immunocompromised or chronically ill children: Lack of knowledge about the vaccine [35], concerns about possible side effects [22,39], and the recent introduction of this vaccine. Fear of disease flare was rarely mentioned, unlike a study of vaccination in lupus patients [22]. Many parents planned to vaccinate their daughters for HPV later. However, HPV vaccination is sometimes associated with a fear of appearing to approve of the onset of sexual relations [42]. The median age at first intercourse in France is 17 years. Therefore, vaccination between 11 and 14 years old could help to separate vaccination from potential sexual disinhibition.

At the time of the study, the nonvalent vaccine had been available for two years, which might explain why only 36% of patients in our cohort received this vaccine.

It might be interesting to conduct this study on young women older than 19 years of age to assess the vaccination rate after catch-up. Similarly, since the generalization of vaccination in boys, this study could be extended to young men with chronic inflammatory diseases.

The findings of this study must be seen in light of some limitations. The cross-sectional design limits the interpretation of the observed statistical associations in terms of causal presumption, while the small size of the study sample leads to low power of the statistical tests performed, limiting the interpretation of the non-significant results. Second, the use of questionnaires is associated with a possible memory bias. We could also discuss the representativeness of our population since all of the patients were followed in expert centers. However, adolescents and young adults with SLE or JIA are intended to be followed in such structures. Finally, although the girls were asked to complete the first questionnaire on their own, the influence of an accompanying adult on their responses, particularly regarding reasons for vaccination or non-vaccination, cannot be excluded.

## 5. Conclusions

Overall, the vaccination coverage in our study is encouraging. Nevertheless, considering the potential high risk for cervical dysplasia, particularly for SLE patients, the coverage remains insufficient. The involvement of health professionals is essential to improve immunization coverage, and strategies should be implemented to provide more accessible information to patients and their parents.

## Figures and Tables

**Figure 1 jcm-11-04137-f001:**
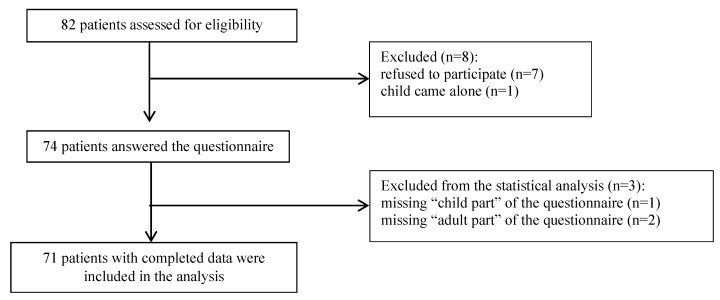
Flowchart of the study participants.

**Figure 2 jcm-11-04137-f002:**
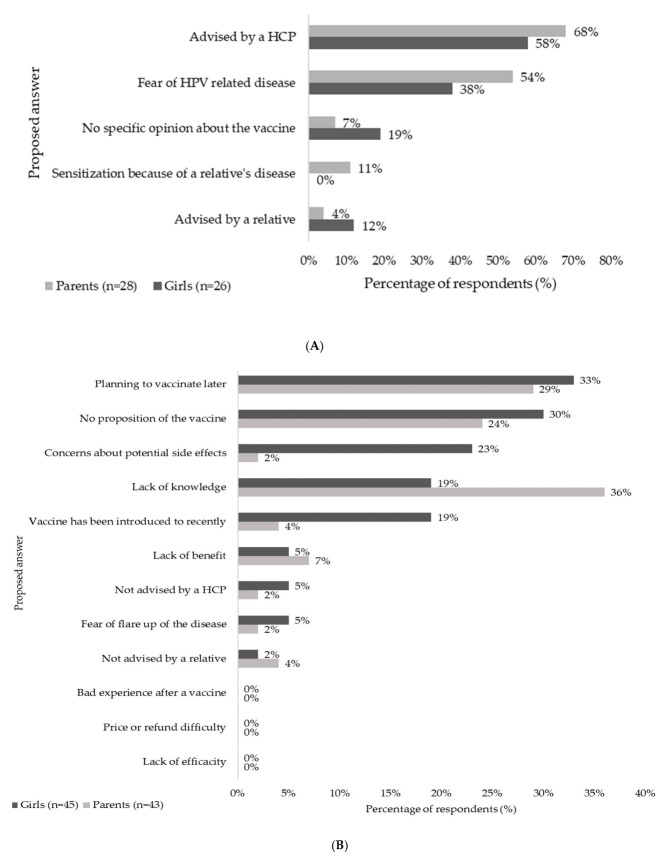
Motivations for and barriers to vaccination. The main motivations (**A**) or barriers (**B**) mentioned by the girls (in dark grey) and their parents (in light grey) for vaccination are shown in this figure. The percentage of respondents is shown on the horizontal axis for each response proposal (on the vertical axis). HCP: Health care professional; HPV: Human papillomavirus.

**Table 1 jcm-11-04137-t001:** Population characteristics.

Characteristics	*n* (%)
Patients	71
Age, years	
Median [Q1; Q3]	13 [12; 15]
Range	11–18
Age category, years	
11–14 (%)	48 (69)
≥15 (%)	22 (31)
Missing data	1 (1)
Relevant pathology	
SLE	16 (23)
JIA	55 (77)
Duration of disease (years)	
Median [Q1; Q3]	7 [4; 11]
Range	1–16
Centre	
HFME (Lyon)	37 (52)
Necker hospital (Paris)	34 (48)
Other vaccines	
Flu	44 (62)
Meningococcus C	31 (44)
Hepatitis B	32 (45)
Pneumococcus	43 (61)
Accompanying Adults	
Father	13 (18)
Mother	58 (82)
Age, years	
Median [Q1; Q3]	46.5 [44; 49.75]
Range	32–58
Educational level	
Primary or lower secondary (ISCED 1–2)	7 (10)
Upper secondary (ISCED 3–4)	37 (52)
Tertiary (ISCED 5–8)	26 (37)
Missing data	1 (1)
History of hepatitis B vaccination	38 (54)

**Table 2 jcm-11-04137-t002:** Characteristics of vaccinated patients (*n* = 28).

Characteristics	*n* (%)
Age (years), Median	15
Relevant pathology	
SLE	7 (25)
JIA	21 (75)
Type of vaccine	
Bivalent	0 (0)
Quadrivalent	17 (61)
Nonvalent	10 (36)
Missing data	1 (3)
Adequate vaccination schedule	
Yes	9 (32)
No	4 (14)
In progress	6 (21)
Missing Data	9 (32)
Age at first injection	
9–10 years old	0 (0)
11–14 years old	12 (43)
≥15 years old	5 (18)
Missing data	11 (39)
Vaccine prescriber	
General practitioner	11 (39)
Hospital pediatrician	11 (39)
Liberal pediatrician	4 (14)
Gynecologist	1 (1)
Missing data	1 (1)

**Table 3 jcm-11-04137-t003:** Comparison of vaccinated and nonvaccinated patients.

Characteristics	Vaccinated Patients (*n* = 28)	Nonvaccinated Patients (*n* = 43)	OR	95% CI	*p* Value *
Patient characteristics					
Age, years					
Median	15	12			<0.001 ^†^
Range	12–18	11–17			
Age Group [11,12]	2 (6%)	30 (94%)	1		<0.001
Age Group [13,18]	26 (68%)	12 (32%)	32.50	6.65–158.80	
Centre					
Paris	18 (53%)	16 (47%)	1		0.025
Lyon	10 (27%)	27 (73%)	0.33	0.12–0.89	
Up to date for other vaccinations					
Flu not-vaccinated	7 (32%)	15 (68%)	1		0.284
Flu vaccinated	20 (45%)	24 (55%)	1.79	0.61–5.24	
Hepatitis B not-vaccinated	5 (22%)	18 (78%)	1		0.009
Hepatitis B vaccinated	18 (56%)	14 (44%)	4.63	1.38–15.56	
Pneumococcus not-vaccinated	2 (40%)	3 (60%)	1		1.000 **
Pneumococcus vaccinated	17 (40%)	26 (60%)	0.98	0.10–12.90	
Meningococcus C not-vaccinated	6 (38%)	10 (62%)	1		0.356
Meningococcus C vaccinated	16 (52%)	15 (48%)	1.78	0.52–6.10	
Relevant pathology					
SLE	7 (44%)	9 (56%)	1		0.698
JIA	21 (38%)	34 (62%)	0.79	0.26–2.45	
Level of drug immunosuppression					
Low	4 (57%)	3 (43%)	1		0.763 ***
Mild	5 (31%)	11 (69%)	0.34	0.05–2.13	
High	19 (41%)	27 (59%)	0.53	0.11–2.63	
Missing data	0	2			
Duration of disease; Mean (SD)	6.7 (3.8)	7.8 (4.1)			0.227 ^†^
Accompanying Adult Characteristics					
Age, years					
Median	45.5 [43.75; 50]	47 [43; 48.75]			0.453 ^†^
Range	38–54	32–58			
Educational level					
Primary or lower secondary (ISCED 1–2)	3 (43%)	4 (57%)	1		0.817 ***
Upper secondary (ISCED 3–4)	15 (41%)	22 (59%)	0.91	0.18–4.66	
Tertiary (ISCED 5–8)	10 (38%)	16 (62%)	0.83	0.15–4.53	
Missing data	0	1			
Hepatitis B not-vaccinated	2 (11%)	16 (89%)	1		0.008
Hepatitis B vaccinated	17 (45%)	21 (55%)	6.48	1.30–32.17	
Mother’s Gynaecological Characteristics					
Regular cervical screening No	6 (50%)	6 (50%)	1		0.336
Regular cervical screening Yes	17 (38%)	28 (62%)	2.20	0.44–11.03	
No previous abnormal cervical smear	20 (43%)	27 (57%)	1		0.237
Previous abnormal cervical smear	2 (22%)	7 (78%)	0.39	0.07–2.06	

* Likelihood ratio test; ^†^ Wilcoxon’s test; ** Fisher’s exact test; *** *p* for trend.

## Data Availability

Data available on request due to restrictions of ethical. The data presented in this study are available on request from the corresponding author.

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
