# Peer review of "Human Papilloma Virus Vaccination in Patients with Rheumatic Diseases in France: A Study of Vaccination Coverage and Drivers of Vaccination"

_jcm, 2022, doi:10.3390/jcm11144137_

Round 1
Reviewer 1 Report
The present study shows interesting results for public health, by evaluating the context of the HPV vaccine in a specific sample. However, the manuscript needs major changes:
- The study has a cross-sectional design, the applicable measure of effect would be the Prevalence Ratio. Why was the Odd Ratio used?
- Was the sample calculation performed? Does the evaluated sample have statistical power?
- In statistical analyses, it is important to evaluate the data distribution using the Kolmogorov-Smirnov and Shapiro-Wilk tests, and then decide on the presentation of the mean or median and standard deviation or interquartile range.
- The multivariate analysis model needs to be further specified. What are the criteria adopted to add the variables in the logistic regression model? Because the Poisson regression model was not used, since it is the most suitable for cross-sectional studies.
- Improve the resolution of images.
Author Response
The authors thanks the reviewer 1 for his comments and questions.
Please see the attachment with point by point response.

Reviewer 2 Report
This study is comparing vaccination rates with the HPV vaccine in two groups of females who also have an underlying autoimmune condition (SLE, JIA). They claim in their conclusions that the rate of vaccination for these females is greater than what is seen from females without these conditions. That said, I did not see any statistical analysis to indicate that 39% was significantly greater than 32.7, which it may be but without age matched controls it is more speculation.
They also indicate that vaccination for HPV seems to be driven by physicians and prior vaccination for HBV vaccination.
What the study does not address, but would be very interesting to know is how does the underlying autoimmune condition affect the overall efficacy of protection. There were some allusions to this early on, but it would have been very interesting to note antibody titers and other immune function assays that could have been performed and compared to an age-matched control group.
Author Response
We thanks the reviewer 2 for his comments.
Please see the attachment with point by point response.

Round 2
Reviewer 1 Report
The authors made the suggested changes.